# OSKAR: Omnimodal Self-supervised Knowledge Abstraction and Representation

**Mohamed Abdelfattah**[*][†]      **Kaouther Messaoud**[*]      **Alexandre Alahi**

École Polytechnique Fédérale de Lausanne (EPFL), Switzerland

`{firstname.lastname}@epfl.ch`

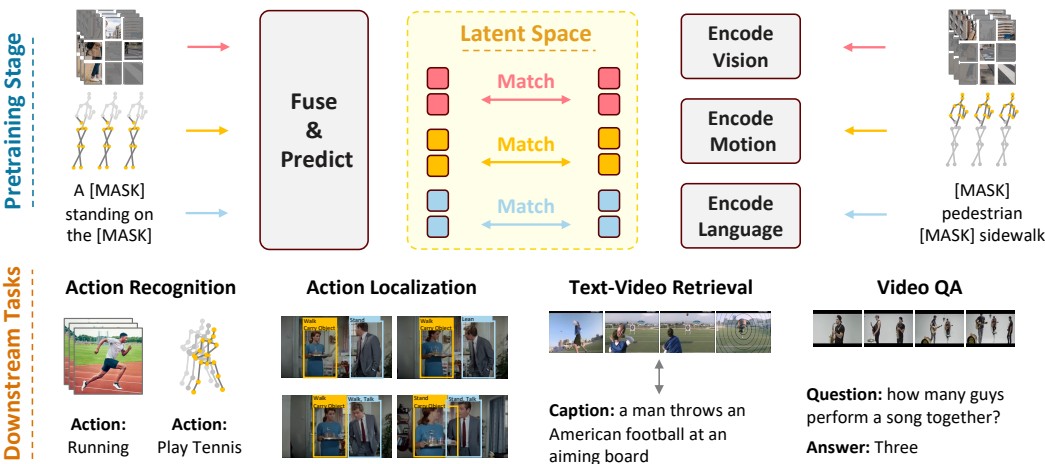

Figure 1: **OSKAR** is a self-supervised multimodal foundation model that learns in the *latent space* using a *fuse-then-predict* strategy. It integrates multiple modalities to capture cross-modal features, matching latent predictions to targets from modality-specific momentum encoders. This preserves uni-modal structure while enabling rich cross-modal learning, and finetuning the unified encoder surpasses specialized models across video, skeleton, and text tasks.

## Abstract

We present OSKAR, the first multimodal foundation model based on bootstrapped latent feature prediction. Unlike generative or contrastive methods, it avoids memorizing unnecessary details (*e.g.,* pixels), and does not require negative pairs, large memory banks, or hand-crafted augmentations. We propose a **novel pretraining strategy**: *given masked tokens from multiple modalities, predict a subset of missing tokens per modality, supervised by **momentum-updated uni-modal target encoders***. This design efficiently utilizes the model capacity in learning high-level representations while retaining modality-specific information. Further, we propose a **scalable design** which decouples the compute cost from the number of modalities using a fixed representative token budget—in both input and target tokens—and introduces a parameter-efficient cross-attention predictor that grounds each prediction in the full multimodal context. We instantiate OSKAR on video, skeleton, and text modalities. Extensive experimental results show that OSKAR's unified pretrained encoder outperforms models with specialized architectures of similar size in action recognition (rgb, skeleton, frozen, low-shot) and localization, video-text retrieval, and video question answering. Project website: https://multimodal-oskar.github.io

---

[*]Equal contribution.

[†]Corresponding author.

39th Conference on Neural Information Processing Systems (NeurIPS 2025).

# 1 Introduction

Human perception is inherently multimodal—we naturally integrate visual, motion, and linguistic cues to form coherent understanding from partial observations. In computer vision, multimodal models offer key advantages: (1) they align with human perception by leveraging visual, structural, and semantic signals; (2) they provide architectural flexibility through unified, reusable representations; and (3) they enhance robustness by fusing complementary inputs (*e.g.,* RGB + LiDAR). Existing multimodal methods typically fall into two categories: *generative* and *contrastive*. Generative approaches [6, 60, 7, 36, 42, 26], often based on masked autoencoding [40], focus on low-level reconstruction, which may waste capacity on irrelevant details. Contrastive methods [34, 83, 3, 55, 88] align high-level embeddings but rely on modality-specific priors, handcrafted augmentations, and lack cross-modal predictive reasoning. Therefore, we ask: *Is it possible to move beyond inefficient reconstruction and restrictive contrastive objectives to learn rich cross-modal representations?*

To address these challenges, we explore multiple strategies for effectively routing multimodal information, culminating in **OSKAR** (**O**mnimodal **S**elf-supervised **K**nowledge **A**bstraction and **R**epresentation), comprising three novel contributions:

**(1) A new pretext task:** *given partial multimodal observations, use cross-modal cues to predict the latent representations of a subset of the missing parts in each modality.* As shown in Fig. 1, OSKAR fuses visible multimodal tokens but infers the missing token representations in each modality separately. This strikes a crucial balance between cross-modal fusion and retention of modality-specific information. Further, by learning in the latent space, the model capacity is efficiently utilized in learning transferrable high-level representations instead of low-level details. Our approach is grounded in the predictive coding theory [65, 30], which posits that the brain learns by predicting internal multimodal representations and minimizing errors, rather than reconstructing raw inputs.

**(2) Modality-specific target encoders:** Unlike prior works [63, 70, 20, 43] distilled from external teachers, OSKAR *trains from scratch* with momentum-updated target encoders—one per modality. These encoders co-evolve with the model and data, generating stable yet adaptive targets that align closely with the model's internal representation space. This design offers an interesting trade-off: with a fixed momentum update rate, we get shared-weight target encoders, thus providing a flexible *modality-agnostic* encoder in fine-tuning; with customized update rates, we allow each modality to evolve at its own learning pace, providing multiple *modality-specific* encoders with peak performance.

**(3) Scalable design:** OSKAR scales efficiently thanks to three key design choices. First, it processes a fixed total number of tokens in *both* the input and target, thus dissociating the compute cost from the number of modalities. Importantly, it ensures fair representation of all modalities, regardless of their raw size, through a Dirichlet allocation strategy. Second, it avoids information leaks through non-overlapping masking, *within* and *between* the inputs and targets. Finally, OSKAR introduces a *unified*, modality-agnostic cross-attention predictor that efficiently anchors predictions in shared multimodal context—seamlessly scaling to new modalities with limited growth in model size.

Though OSKAR supports plug-and-play extensibility, we instantiate it on three distinct modalities—video (dense), poses (sparse), and text (symbolic)—forming a challenging testbed for evaluation. Trained *entirely with pseudo-labels* and without manual annotations, OSKAR matches or surpasses specialized models on RGB- and skeleton-based action recognition (86.1% K400, 91.1% NTU120-XSub), spatiotemporal action localization (37.9 mAP AVA), text-video retrieval (50.4 R@1 MSRVTT), and video question answering (49.3% MSRVTT-QA). It also outperforms baselines in low-sample, low-parameter, and low-label settings—highlighting its efficiency and versatility.

# 2 Related Works

**Generative architectures (GAs).** Generative self-supervised models corrupt the input and train an encoder together with a lightweight reconstruction head to in-paint the missing content in *input space*. MAE [40] restores masked image patches; VideoMAE [71] extends the idea to spatio-temporal tubes; OmniMAE [35] accepts mixed image–video inputs; Other unified masked models[78, 68] extend it to vision–language and follow-ups [6, 60, 7, 33] broaden the paradigm to additional modalities. While effective at capturing low-level details, these methods optimize pixel-level losses, often diverting model capacity toward reconstructing semantically uninformative elements like texture or lighting.

**Joint-embedding architectures (JEAs).** JEAs learn by *aligning* representations. Given two or more views of the *same* instance—obtained through data augmentation or drawn from another modality—the model is trained with a contrastive [19, 64], redundancy-reduction [89, 11] or average embedding entropy maximization [15, 4, 38] objective that pulls similar pairs together in feature space while pushing dissimilar ones apart. In cross modal context, CLIP [64] aligns whole-image and sentence embeddings via a pure contrastive loss; More recent models [50, 23, 61, 46] add a momentum teacher that *self-distills* latent knowledge to the student. ImageBind [34] generalises the recipe to six sensory streams. The objective enforces only *global* agreement between paired embeddings. As a result, JEAs excel at zero-shot recognition and retrieval, yet they lack a mechanism for *structured* cross-modal prediction—for instance, nothing in CLIP compels the model to infer a missing video patch from a co-occurring caption.

**Joint-embedding predictive architectures (JEPAs).** JEPAs [47] blend both worlds by replacing pixel reconstruction with *latent* prediction. A predictor maps visible-context embeddings to the target embeddings of masked regions; the loss is computed in feature space. Hence, the objective is to *learn representations that are predictive of each other*. On images, I-JEPA [5] demonstrates that latent prediction yields strong abstraction with lower compute than MAE [40]. V-JEPA [12] and S-JEPA [1] adapt the idea to video and skeleton data, respectively. iBOT [91] and data2vec [8, 9] also fall under the JEPA framework, performing latent prediction from masked inputs to match teacher representations—at the patch level in iBOT, and in a modality-agnostic manner in data2vec.

**OSKAR diverges from these methods in the following:** (1) *Cross-modal latent prediction:* Compared to [5, 12, 1], OSKAR *intentionally* uses far fewer per-modality tokens; yet, it complements them with cross-modal cues. Hence, the model is forced to learn from fused cues from **all** modalities to predict the missing latent features in each. (2) *Modality-specific target encoders:* OSKAR employs separate target encoders per modality, balancing intra-modality structure with cross-modal alignment, and allowing a flexible design choice between shared or customized encoders. (3) *Scalable cross-modal masking:* OSKAR masks both the inputs and targets with fixed budgets to keep the compute cost manageable with increasing modalities. Further, it uses Dirichlet sampling to process dense and sparse modalities fairly, while ensuring cross-modal exclusivity to avoid trivial shortcuts.

## 3 Methodology

**Overview.** OSKAR operates in two stages: pretraining and fine-tuning. In pretraining (see Fig. 2), OSKAR optimizes a novel "fuse–then–predict" task: given partially masked multimodal tokens, it fuses visible inputs via a cross-modal fusion encoder and predicts modality-specific *latent representations* for masked tokens using shared-weight per-modality predictors. Target representations are generated by momentum-updated encoders, providing stable supervision. Importantly, *learning occurs entirely in the latent space*, emphasizing high-level semantics over low-level reconstruction. In fine-tuning, the target encoders are adapted directly to downstream tasks. All backbones are standard transformers [24], enabling seamless integration into existing transformer pipelines.

### 3.1 Architecture

**Tokenization and Embedding.** Following [60, 7], we first tokenize raw inputs (*e.g.,* videos, skeletons, text) using modality-specific encoders [12, 92, 69]; tokenization serves only to accelerate training and is not required for the method itself, producing tokens $\mathbf{T}_m$ for each modality $m$. These are linearly projected via $g_m$ into a shared embedding space $\mathbf{E}_m \in \mathbb{R}^{N_m \times d}$ of $N_m$ modality tokens, enabling a unified encoder across modalities without task-specific customizations. Each $\mathbf{E}$ is then augmented with learnable positional $e_{(\text{pos})}$, modality $e_{(\text{mod})}$, and auxiliary $e_{(\text{aux})}$ signals. In particular, $e_{(\text{aux})}$ are modality-specific auxiliary signals proposed to resolve ambiguities (*e.g.*, helping the predictor establish pixel-to-keypoint correspondence and disambiguate multiple people in a scene. Without them, the model may associate skeletons with the wrong subject, especially in crowded scenes). We then apply our masking strategy (explained later) to keep only a few non-overlapping patches in the fusion and target encoders, denoted $\mathbf{E}^f \in \mathbb{R}^{N^f \times d}$ and $\mathbf{E}^t \in \mathbb{R}^{N^t \times d}$, where $N^f$ and $N^t$ denote the number of input and target patches, respectively.

**Fusion encoder.** Given $\mathbf{E}^f$, the fusion encoder's objective is to exploit the complementary cues from *all* modalities to infer the missing information in each. To that end, we feed $\mathbf{E}^f$ to the fusion

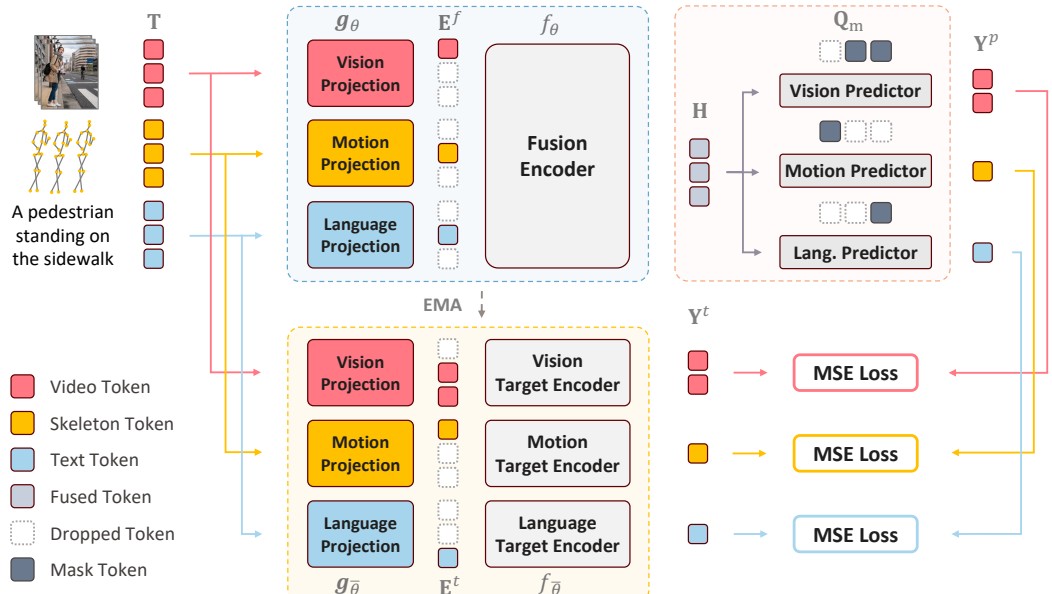

Figure 2: **OSKAR Architecture.** Multimodal tokens (*e.g.,* video, skeleton, text) are projected into a shared space and split into two branches: (1) a fusion encoder processes visible tokens to produce fused representations; (2) modality-specific target encoders generate target embeddings. A predictor estimates masked representations from fused tokens, supervised via MSE loss against targets. Target encoders are updated via EMA of the fusion encoder and are used exclusively in fine-tuning.

transformer $f_\theta$, where all modality tokens interact with each other through the inter-modal Multi-Head Self-Attention (MHSA) mechanism [72], yielding fused representations $\mathbf{H} \in \mathbb{R}^{N^f \times d}$.

**Predictor.** The role of the predictor is to generate the latent features for a subset of the missing tokens in every modality. These predictions are conditioned on predictor-specific *target-location* queries $\mathbf{Q}^t \in \mathbb{R}^{N^t \times d}$, which are learnable mask tokens $\mathbf{M}^t$, augmented with positional $e^p_{(\text{pos})}$, modality $e^p_{(\text{mod})}$, and auxiliary cues $e^p_{(\text{aux})}$. The predictor processes each set of queries $\mathbf{Q}^t_m$ of modality $m$ through a transformer with alternating self-attention and cross-attention layers. Self-attention allows queries to share information and capture intra-modality structure, while cross-attention integrates information from the fused representations $\mathbf{H}$. At each cross-attention layer, $\mathbf{Q}^t_m$ attend to all tokens in $\mathbf{H}$:

$$\mathbf{Y}^p_m = \text{MHCA}(\mathbf{Q}^t_m, \mathbf{H}, \mathbf{H}),\tag{1}$$

where $\mathbf{Y}^p_m$ are the predicted representations and MHCA is Multi-Head Cross-Attention. Hence, $\mathbf{Y}^p \in \mathbb{R}^{N^t \times d}$ concatenates all per-modality predictions $\mathbf{Y}^p_m$. Importantly, there is a single predictor network with shared weights across modalities. However, it operates like several—one per modality—while retaining the efficiency and regularization benefits of shared weights. This design (i) allows queries to exchange information within a modality (via self-attention), (ii) grounds every prediction in the full multimodal context $\mathbf{H}$ (via cross-attention), and (iii) adapts to new modalities without additional heads, all while remaining decoupled from the fusion encoder for easy transfer to downstream tasks.

**Target encoders.** Unlike raw inputs (*e.g.,* pose joints)—often subtle, noisy, and isolated—the target encoders provide clean, high-level targets $\mathbf{Y}^t \in \mathbb{R}^{N^t \times d}$, steering the model away from overfitting to spurious details. A key innovation in OSKAR is the use of *modality-specific target encoders*—rather than a single cross-modal target encoder—to balance two objectives: enabling the fusion encoder to learn cross-modal abstractions while preserving each modality's structure and information content. While the fusion encoder and predictor parameters $(\theta, \vartheta)$ are updated with gradients, each target encoder's parameters $\bar{\theta}_m$ are updated via an exponential moving average (EMA) of $\theta$:

$$\bar{\theta}_m \leftarrow \lambda_m \bar{\theta}_m + (1 - \lambda_m)\theta,\tag{2}$$

where $\lambda_m$ is a modality-specific momentum coefficient. OSKAR supports two target encoder update strategies: (1) *Shared-weight target encoders* (*i.e.,* same $\lambda$ for all $m$): offering a unified target encoder

network with strong downstream multimodal performance, and (2) *Customized target encoders* (*i.e.,* modality-specific $\lambda_m$ values) offering multiple target encoders with better uni-modal performance, trained with varying update rates that accommodate each modality's learning dynamics. For flexible evaluation, we adopt the first option by default but we study all design choices in ablations sec 5.

## 3.2 Training

**Pretraining: Fuse–then–Predict.** Given partially masked multimodal tokens, OSKAR fuses visible inputs with a cross-modal transformer and predicts modality-specific *latent* features for masked tokens via a single shared predictor. Targets are produced by momentum-updated encoders for stable supervision. Learning occurs entirely in latent space, emphasizing high-level semantics over low-level reconstruction.

**Pretraining Objective.** With both $\mathbf{Y}^p$ and $\mathbf{Y}^t$ now computed, we optimize OSKAR by minimizing a Mean Square Error (MSE) loss between the predicted and target representations:

$$\mathcal{L} = \frac{1}{N^t} \sum_{i \in N^t} \left\| \mathbf{Y}_i^p - \mathbf{Y}_i^t \right\|_2^2. \tag{3}$$

Crucially, by predicting in feature space, we bypass the need for task/modality-specific losses (*e.g.,* pixel-wise MSE for images or cross-entropy for text). This grants **universal flexibility**: The shared latent objective delivers comparable gradients across modalities, simplifying optimization, reducing negative transfer, and supporting graceful scaling to new modalities without the loss-balancing game.

**Cross-Modal Masking Strategy.** Multimodality introduces three core challenges: scalability and efficiency with increasing modalities, imbalance because of modality size disparities, and trivial cross-modal shortcuts. OSKAR addresses them all with key design choices: (1) *Fixed token budget:* Inspired by [6], we sample a fixed total budget of $N$ tokens, but for *both* inputs and targets, decoupling the compute cost from the number of input modalities. (2) *Adaptive budgeting:* Instead of naive random sampling—which would let larger modalities (*e.g.,* video) overwhelm smaller ones (*e.g.,* text)—we allocate a fraction $r_m$ of $N$ to each modality $m$ by drawing $r_m$ from a symmetric Dirichlet($\alpha$) distribution, ensuring $\sum_m r_m = 1$. Lower $\alpha$ values (*e.g.,* 0.1–0.5) assign all $N$ to a single modality, while higher values ($\geq 1$) promote more balanced allocations. (3) *Cross-modal exclusivity:* We sample $N_m = r_m \times N$ tokens per modality under a cross-modal spatio-temporal exclusivity constraint, which prevents trivial prediction—if a video patch is visible to the fusion encoder, the corresponding skeleton joint is masked from the target encoder. To reduce redundancy in sequence modalities, we mask contiguous spatio-temporal tubes, encouraging the model to reason over extended, coherent structures rather than isolated tokens. The fusion and target encoder token sets, $\mathbf{X}^f$ and $\mathbf{X}^t$, are sampled independently and are strictly disjoint ($\mathbf{X}^f \cap \mathbf{X}^t = \emptyset$), preventing direct copying and enforcing meaningful prediction of unseen information. This strategy generalizes seamlessly to new modalities while maintaining computational fairness and discouraging shortcuts.

**Fine-Tuning.** After pretraining, the target encoders are adapted directly to downstream tasks. Because all backbones are standard transformers, fine-tuning integrates seamlessly into existing transformer pipelines.

## 4 Experimental Results

### 4.1 Pretraining Details.

**Datasets.** We train OSKAR **entirely with pseudo-labels** on 10M videos from OpenHumanVid[3] [49] (13.2M videos, 16.7K hours). Using YOLO11 [44], we pseudo-label pose, tracking, and detection, selecting top-3 individuals in crowded videos via visibility, motion, keypoint/bbox confidence, and center proximity. Captions are generated with MiniCPM [87] and CogVLM [41]. Following [60, 7], we speed up processing by pre-tokenizing the videos, skeletons, and text using V-JEPA [12], MotionBERT [92], and WordPiece [69], respectively. Downstream benchmarks include: Kinetics-400 [45] and Something-Something V2 [37] (RGB action recognition), NTU60 [66] and NTU120 [54] (skeleton action recognition), AVA [39] (action localization), MSRVTT [84]/MSVD [16]/VATEX [79] (text-video retrieval), and MSRVTT-QA [84]/MSVD-QA [82]/TGIF-FrameQA [52] (videoQA).

---

[3]At the submission time of this paper, only 10M videos from OpenHumanVid were publicly released.

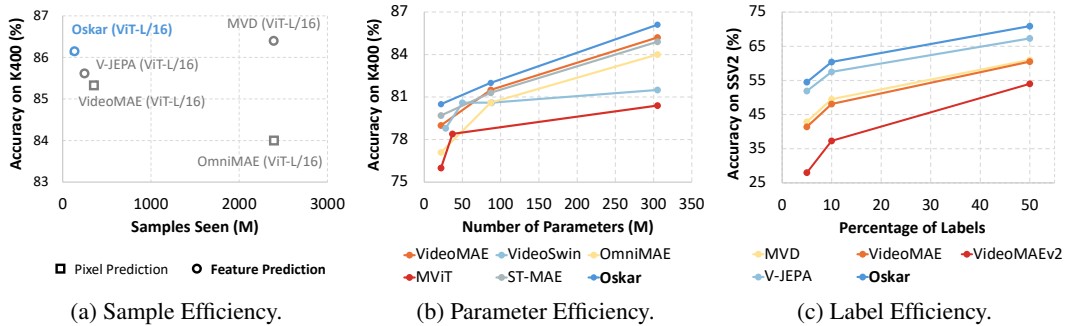

(a) Sample Efficiency.     (b) Parameter Efficiency.     (c) Label Efficiency.

Figure 3: **OSKAR exhibits strong scalability** with (a) fewer samples, (b) less parameters, and (c) less labels per class than comparable methods.

**Pre-training.** We use standard ViT-S, ViT-B, and ViT-L [24] backbones with learnable positional encodings. All target encoders use a shared EMA update parameter ($\lambda = 0.998$). Models are randomly initialized and trained on 500B tokens (10B warmup) using AdamW [58] ($\beta_1 = 0.9$, $\beta_2 = 0.95$), a base learning rate of 1e-4, cosine decay, batch size 8192, and weight decay 0.05. Transformers use SwiGLU [67] activations and bfloat16 [14] precision. Each model processes $N^s = N^t = 128$ tokens per step, with aggressive masking (<5%, 128 of 2640 tokens visible) via non-overlapping modality masks and Dirichlet-sampled allocation ratios ($\alpha = 0.5$). Pretraining uses 256 GH200 GPUs. Video inputs are 16 frames (stride 2), resized to $224 \times 224$; skeletons are temporally aligned and normalized to match video dimensions.

## 4.2 Main Results

By default, OSKAR is fine-tuned per task using standard inputs (*e.g.,* video only for action recognition).

**Action recognition.** OSKAR consistently outperforms specialized video-only models across multiple model sizes and datasets without extra cost. On K400 (Tab. 1), OSKAR outperforms V-JEPA [12] and VideoMAE [71] with the same ViT-L backbone, and scales better with limited data (Fig. 3a) and fewer parameters (Fig. 3b). It also beats MViT [27], BEVT [77], and TimeSformer [13] with fewer frames/parameters. For SSv2, gains reach **+3.3%** over VideoMAE (ViT-S) and **+2.5%** over V-JEPA (ViT-L). Beyond video, OSKAR transfers effectively to skeleton-based action recognition: on NTU60 and NTU120 XSub (Tab. 4), OSKAR-B achieves new state-of-the-art results, outperforming MotionBERT [92] by **+0.9%** and **+6.1%**, respectively.

**Frozen low-shot action recognition.** OSKAR's frozen features deliver strong out-of-the-box performance on SSV2 without fine-tuning, particularly in label-scarce settings (Fig. 3c). With only 5%, 10%, or 50% of the labels, it consistently outperforms other models, achieving a **26.5%** absolute gain over VideoMAEv2 [75] and **2.6%** over V-JEPA [12] at the 5% setting. Unlike V-JEPA's visual-only pretraining, OSKAR *intentionally* reduces the number of visible video tokens but supplements them with motion and language tokens to ground the visual features in semantic and structural information, promoting generalization and yielding consistent gains (**+2.6–3.6%**) across all low-shot settings.

**Spatiotemporal action localization.** OSKAR transfers effectively to spatiotemporal action localization, consistently outperforming larger video-only models. With just 22M parameters (ViT-S), it matches SlowFast [28] (59M) and surpasses VideoMAE-S by **+5.0** mAP. Scaling up, OSKAR outperforms VideoMAE-B by **+3.5** mAP (ViT-B) and, at the large scale, exceeds V-JEPA and VideoMAE-H by **+1.7** and **+1.4** mAP, respectively—while using less than half the parameters of VideoMAE-H. These gains reflect OSKAR's ability to leverage multimodal pretraining to capture both semantic context and fine-grained motion cues, essential for localizing actions in space and time.

**Text-video retrieval.** Compared to methods trained with <200M pairs, OSKAR (ViT-L) achieves 50.4 R1 on MSRVTT (**+2.6** over OmniVL [73]); 54.4 R1 on MSVD (**+4.3** over LAVENDER [51]); 54.1 R1 on VATEX (**+3.7** over CLIP4CLIP [59]). Notably, OSKAR performs within a close margin to specialist models trained with 2–3× more data (*e.g.,* only **1.0** below Slide4Video on MSR-VTT) and excels in video-to-text retrieval (**+1.6** to **+5.4** R@1 over CLIP2TV [32]/CenterCLIP [90]/CLIP4Clip [59].

Table 1: **RGB-based action recognition** accuracy (%) on Kinetics-400 [45].

| Method | Resolution | GFLOPs | Acc. |
|---|---|---|---|
| *Small Models (<80M parameters)* | | | |
| VideoMAE-S [71] | 16×224² | 57 | 79.0 |
| SlowFast+NL [28] | 80×224² | 234 | 79.8 |
| MViTv1-B [27] | 32×224² | 170 | 80.2 |
| **OSKAR-S** | 16×224² | 57 | **80.5** |
| *Medium Models (80-150M parameters)* | | | |
| OmniMAE-B [35] | 16×224² | 180 | 80.6 |
| TimeSformer-B [13] | 96×224² | 2380 | 80.7 |
| BEVT-B [77] | 32×224² | 282 | 81.1 |
| ST-MAE-B [29] | 16×224² | 180 | 81.3 |
| VideoMAE-B [71] | 16×224² | 180 | 81.5 |
| **OSKAR-B** | 16×224² | 180 | **82.0** |
| *Large Models (150-700M parameters)* | | | |
| VideoSwin-L [57] | 32×224² | 604 | 83.1 |
| OmniMAE-L [35] | 16×224² | 597 | 84.0 |
| VideoMAE-L [71] | 16×224² | 597 | 85.2 |
| V-JEPA-L [12] | 16×224² | 597 | 85.6 |
| **OSKAR-L** | 16×224² | 596 | **86.1** |

Table 2: **Action detection** mAP on AVA v2.2 [39], all using 16×224² resolution.

| Method | PT data | Param (M) | mAP |
|---|---|---|---|
| *Small Models (<80M parameters)* | | | |
| VideoMAE-S [71] | K400 | 22 | 22.5 |
| MViTv1-B [27] | K600 | 36.3 | 26.1 |
| MViTv2-B [53] | K400 | 34.5 | 26.2 |
| SlowFast [28] | K600 | 59.2 | **27.5** |
| **OSKAR-S** | OpenHumanVid | **22** | **27.5** |
| *Medium Models (80-150M parameters)* | | | |
| VideoMAE-B [71] | K400 | 87 | 26.7 |
| **OSKAR-B** | OpenHumanVid | 87 | **30.2** |
| *Large Models (150-700M parameters)* | | | |
| VideoMAE-L [71] | K400 | 305 | 34.3 |
| ST-MAE-L [29] | K400 | 304 | 34.8 |
| VideoMAE-H [71] | K400 | 633 | 36.5 |
| V-JEPA-L [12] | VideoMix2M | 200 | 36.2 |
| ST-MAE-L [29] | K700 | 304 | 37.3 |
| **OSKAR-L** | OpenHumanVid | 305 | **37.9** |

Table 3: **Text-to-video retrieval** Recall@1 on MSRVTT [84], MSVD [16], and VATEX [79].

| Method | Pairs (M) | MSRVTT | MSVD | VATEX |
|---|---|---|---|---|
| *Methods using large-scale data (>400M samples)* | | | | |
| Cap4Video [81] | 400 | 51.4 | 51.8 | 66.6 |
| S4Vid-L[86] | 400 | 51.4 | 54.9 | **67.9** |
| CLIP-ViP [85] | 500 | 54.2 | - | - |
| IntVideo [80] | 646 | **55.2** | **58.4** | - |
| *Methods using < 200M samples* | | | | |
| TeachText [21] | - | 29.6 | 25.4 | 53.2 |
| CLIP4CLIP [59] | 100 | 46.2 | - | 50.4 |
| Frozen [10] | 5 | 31.0 | 33.7 | - |
| VIOLET [31] | 138 | 34.5 | - | - |
| SUPPORT [62] | 100 | 30.1 | 28.4 | 45.9 |
| LAV. [51] | 30 | 40.7 | 50.1 | - |
| Singularity [48] | 17 | 42.7 | - | - |
| UMT-B [56] | 5 | 46.3 | 47.4 | - |
| DRL-B [76] | - | 47.6 | 47.0 | 44.6 |
| OmniVL [73] | 17 | 47.8 | - | - |
| OSKAR-S | 160 | 45.5 | 47.5 | 49.4 |
| OSKAR-B | 160 | 50.1 | 52.4 | 50.1 |
| **OSKAR-L** | **160** | 50.4 | 54.4 | 54.1 |

Table 4: **Skeleton-based action recognition** accuracy (%) on NTU60 [66] and NTU120 [54].

| Method | Param. (M) | NTU60 XSub | NTU60 XView | NTU120 XSub | NTU120 XSet |
|---|---|---|---|---|---|
| MoBERT [92] | 62 | 93.0 | 97.2 | 84.8 | 86.4 |
| S-JEPA [1] | 21 | 93.1 | 97.6 | 90.3 | 91.3 |
| MaskCLR [2] | 62 | 93.9 | 97.3 | 87.4 | 89.5 |
| PC3D [25] | 2 | 94.1 | 97.1 | 86.9 | 90.3 |
| OSKAR-S | 22 | 93.7 | 97.3 | 89.6 | 89.6 |
| OSKAR-B | 86 | 93.9 | 97.3 | 90.9 | **92.2** |
| **OSKAR-L** | 305 | **94.3** | **97.8** | **91.1** | 92.0 |

Table 5: **VideoQA** accuracy (%) on MSRVTT-QA [82], MSVD-QA [16], and TGIF [52].

| Method | Pairs (M) | MSRVTT | MSVD | TGIF |
|---|---|---|---|---|
| IntVideo [80] | 646 | 47.1 | 55.5 | 72.2 |
| GIT2 [74] | 12900 | 45.6 | 58.2 | 74.9 |
| VALOR-L [17] | 433 | 49.2 | **60.0** | 78.7 |
| COSA [18] | 415 | 49.2 | **60.0** | **79.5** |
| OSKAR-S | 160 | 46.7 | 56.8 | 73.2 |
| OSKAR-B | 160 | 48.9 | 58.3 | 76.1 |
| OSKAR-L | 160 | **49.3** | 59.7 | 79.0 |

Its multimodal pretraining—without external teachers or massive data—outperforms specialized retrieval models of comparable data-parameter scale.

**Open-ended Video Question Answering.** Without QA-specific architecture modifications, OSKAR demonstrates strong gains on VideoQA benchmarks across two configurations: (1) using OSKAR's visual encoder with a BERT [22] text encoder, and (2) using OSKAR for both video and text encoding. On MSRVTT-QA, OSKAR-L outperforms InternVideo [80] by **+2.2** points and performs on par with VALOR-L [17] (+0.1). On MSVD-QA, it is on par with COSA [18] (–0.3) while surpassing InternVideo by **+4.2**. On TGIF, OSKAR-L performs competitively, coming within 0.5 points of COSA. These results highlight OSKAR's effective general-purpose representations for QA tasks, even with ∼2–4× fewer training pairs.

**Qualitative Results.** To assess OSKAR's cross-modal predictions, we freeze the pretrained fusion encoder and predictor, and train a lightweight transformer decoder to map features to joint-space coordinates. As shown in Fig. 4, OSKAR accurately reconstructs missing human poses from video and text tokens. Notably, the first column shows that bounding box embeddings effectively guide pose prediction, even in cluttered scenes with multiple people. These results highlight OSKAR's strong multimodal grounding and ability to preserve spatial structure.

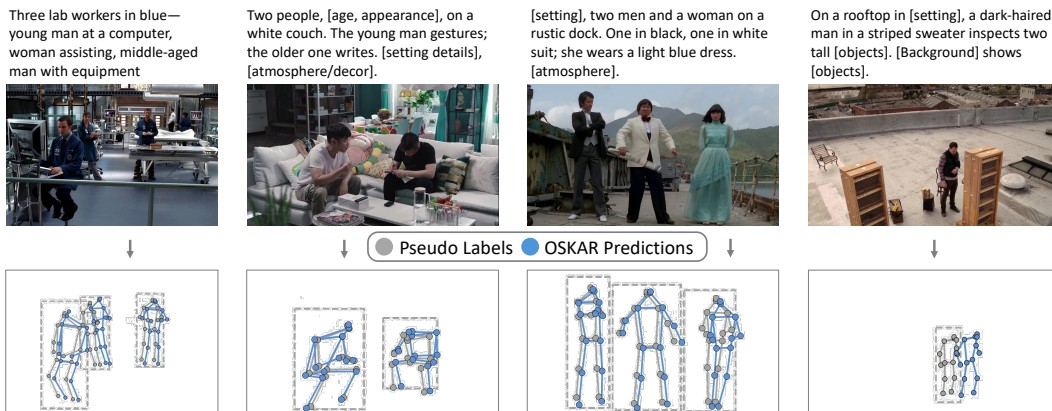

Three lab workers in blue—young man at a computer, woman assisting, middle-aged man with equipment

Two people, [age, appearance], on a white couch. The young man gestures; the older one writes. [setting details], [atmosphere/decor].

[setting], two men and a woman on a rustic dock. One in black, one in white suit; she wears a light blue dress. [atmosphere].

On a rooftop in [setting], a dark-haired man in a striped sweater inspects two tall [objects]. [Background] shows [objects].

Pseudo Labels    OSKAR Predictions

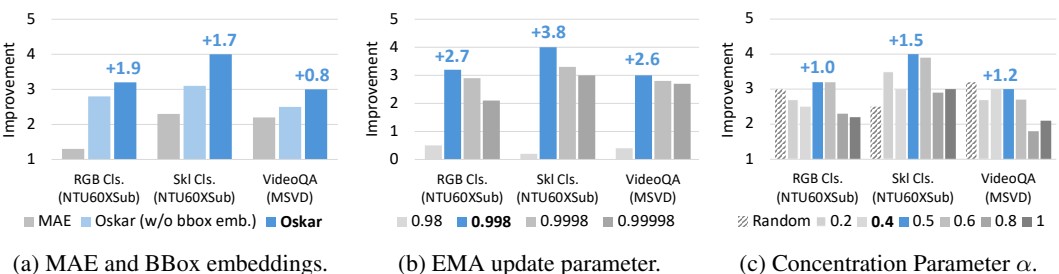

Figure 4: **Visualization of predicted pose features.** OSKAR accurately predicts human poses from video and text, guided by bounding boxes, even in cluttered, multi-person scenes.

(a) MAE and BBox embeddings.  MAE  Oskar (w/o bbox emb.)  **Oskar**

(b) EMA update parameter.  0.98  **0.998**  0.9998  0.99998

(c) Concentration Parameter $\alpha$.  Random  0.2  **0.4**  0.5  0.6  0.8  1

Figure 5: Ablations on (a) MAE and bounding box embeddings, (b) the EMA update parameter, and (c) the $\alpha$ value of the Dirichlet distribution. Blue denotes the default setting of OSKAR. **Blue bold** numbers indicate the difference between our default setting and lowest bar.

# 5   Ablation Studies

Before large-scale pre-training, we ablate design choices by pre-training ViT-S on 100K OpenHumanVid samples, then fine-tuning on NTU60-XSub [66] for RGB (VidCls) and skeleton (SklCls) action recognition, and MSVD [16] for VidQA, comparing to training from scratch (baseline).

**Effect of adding modalities during pre-training.** Table 6 shows that combining modalities consistently boosts performance across tasks. Pretraining with video or skeleton alone yields moderate gains (*e.g.,* **+2.6** for VidCls, **+2.9** for SklCls). Adding text with video further improves VidQA results (**+2.1**), underscoring its value for semantic understanding. The best performance (**+3.2/+4.0/+4.3** for VidCls/SklCls/VidQA) comes from using all three modalities, confirming that multimodal grounding of appearance, motion, and language produces more transferable features.

**Staged Multimodal Attention Routing.** Ablations on OSKAR's attention routing (Table 8) show that the best performance (+3.2 VidCls, +4.0 SklCls, +4.3 VidQA) comes from using this configuration: cross-attention in the fusion encoder to align complementary signals early (*e.g.,* motion and pose), enabling richer representations; intra-modality attention in the target encoder to preserve modality-specific structure, yielding clearer supervision; and the predictor's hybrid setup—first self-attention within modalities, then cross-attention to fused features—balances specialization with contextual grounding (+0.1/-0.3/+0.5 over full self-attention). This staged strategy, integrating early and specializing late, reflects how humans process and combine sensory inputs.

**Prediction in the input vs feature space.** Figure 5a shows that predicting in feature space outperforms input-space reconstruction (MAE) by +1.9 VidCls, +1.7 SklCls, and +0.8 VidQA. These gains echo the advantages of self-distillation reported in unimodal models [5, 12, 15, 38]. Whereas MAE spends capacity reproducing low-level artefacts such as blur or illumination, OSKAR concentrates on semantic cues—e.g., motion dynamics—and its momentum-updated target encoders further stabilise the supervision signal.

Table 6: **Impact of adding video, skeleton, and text modalities during pre-training.**

| Video | Skeleton | Text | VidCls | SklCls | VidQA |
|---|---|---|---|---|---|
| | Baseline | | 88.2 | 75.7 | 39.3 |
| ✓ | | | +2.6 | – | – |
| ✓ | | ✓ | +2.2 | – | +2.1 |
| | ✓ | | – | +2.9 | – |
| | ✓ | ✓ | – | +3.0 | – |
| ✓ | ✓ | | +2.5 | **+4.1** | – |
| ✓ | ✓ | ✓ | **+3.2** | +4.0 | **+4.3** |

Table 7: **Shared vs customized target encoders.** Update speeds: + (slow, $\lambda = 0.99998$), ++ (moderate, $\lambda = 0.9998$), +++ (fast, $\lambda = 0.998$).

| Video | Skeleton | Text | VidCls | SklCls | VidQA |
|---|---|---|---|---|---|
| | Baseline | | 88.2 | 75.7 | 39.3 |
| + | + | + | +3.2 | +4.0 | +4.3 |
| +++ | + | ++ | +3.2 | +3.9 | +3.9 |
| + | +++ | ++ | +2.6 | +2.9 | +3.7 |
| ++ | + | +++ | +3.2 | +3.6 | +3.8 |
| + | ++ | +++ | +2.8 | +3.9 | +4.0 |
| +++ | ++ | + | **+5.0** | **+5.4** | **+5.2** |
| ++ | +++ | + | +4.5 | +4.8 | +5.0 |

Table 8: **Ablation on modality attention routing.** "S": separate; "C": cross-modality.

| Fusion | Target | Pred. | VidCls | SklCls | VidQA |
|---|---|---|---|---|---|
| | Baseline | | 88.2 | 75.7 | 39.3 |
| S | S | S | +1.1 | +3.4 | +2.8 |
| S | S | C | +1.3 | +2.9 | +3.2 |
| S | C | S | +0.8 | +2.8 | +3.4 |
| C | C | C | +1.2 | +2.2 | +3.3 |
| C | C | S | +1.0 | +2.3 | +3.2 |
| **C** | **S** | **S** | **+3.2** | **+4.0** | **+4.3** |

Table 9: **Impact of the number of input and target tokens.**

| Input | Target | VidCls | SklCls | VidQA |
|---|---|---|---|---|
| Baseline | | 88.2 | 75.7 | 39.3 |
| 64 | 64 | +1.0 | +2.9 | +2.7 |
| 128 | 128 | +3.2 | +4.0 | +4.3 |
| 256 | 256 | +3.3 | +3.3 | +3.6 |
| 128 | 256 | +3.1 | +4.0 | +4.2 |
| **64** | **256** | **+3.4** | **+4.3** | **+4.6** |

**Shared-weight vs Customized target encoders.** The EMA coefficient $\lambda$ determines how quickly the target encoder tracks the fusion encoder. At $\lambda = 0$, the encoders are identical, causing representation collapse and near-random performance. Too small a $\lambda$ (*e.g.,* 0.1), i.e., fast updates, destabilizes training, while too large a value (*e.g.,* 0.99998) prevents the target from adapting to new updates. We evaluate two target encoder variants under this trade-off. *(i) Shared-weight target encoders:* A single encoder with a global $\lambda$ performs best at $\lambda = 0.998$ (+2.7 VidCls, +3.8 SklCls, +2.6 VidQA; Fig.5b). *(ii) Customized target encoders:* Assigning modality-specific $\lambda$ values—fast for video (0.998), moderate for skeleton (0.9998), and slow for text (0.99998)—yields the best overall results (+5.0 VidCls, +5.4 SklCls, +5.2 VidQA; Table7). This asymmetry aligns with each modality's nature: videos (low variability, high redundancy) benefit from fast updates to focus on motion; skeletons (moderate, structured variation) require balanced updates; and text (high variability, discrete tokens) improves with slower updates to preserve semantic consistency.

**Input and target number of tokens.** Table 9 shows that more tokens generally improve performance, with the best results at 64 input / 256 target tokens. A 128/128 setting retains 95% of the gains while using less than half the compute, and is thus adopted as default.

**Controlling Modality Mix.** We analyze the impact of the Dirichlet concentration parameter $\alpha$ on modality token sampling (Fig. 5c). Low $\alpha$ skews sampling toward a single modality, while high $\alpha$ enforces uniformity. Setting $\alpha = 0.5$ yields the best trade-off (+0.2 VidCls, +1.5 SklCls, -0.2 VidQA vs. random). Random sampling favors video due to its token volume, benefiting video tasks, but Dirichlet sampling ensures balanced modality representation, improving SklCls and maintaining competitive video performance. We adopt $\alpha = 0.5$ to promote balanced, modality-aware training.

**Bounding box embeddings.** Adding bounding box embeddings improves performance across tasks as illustrated in Fig. 5a by providing spatial cues for person-specific predictions. Removing them causes some ambiguity in multi-person settings. These consistent gains highlight the value of simple spatial priors for learning more discriminative representations.

# 6  Conclusion and limitations

We introduced OSKAR, a novel paradigm for multimodal self-supervised learning that learns semantically rich representations via latent feature prediction. OSKAR introduces a *fuse-then-predict* pretext task, modality-specific momentum encoders for stable supervision, and a scalable masking strategy for balanced and efficient learning. Trained across video, skeleton, and text, OSKAR outperforms specialized models on diverse downstream tasks, while remaining efficient and label-agnostic.

Its modular design supports extensions to new modalities, larger datasets, and adaptive learning dynamics, offering a strong foundation for future multimodal research.

While OSKAR establishes a new state of the art, some limitations offer promising directions for further improvement: (1) *Expanding Modalities:* Although OSKAR currently integrates video, skeleton, and text, adding additional modalities (*e.g.,* audio, depth, IMU) could further enrich the learned representations and unlock new applications. (2) *Scaling Data:* Pretraining on even larger and more diverse datasets would likely enhance the model's generalization and transferability, particularly for complex multimodal reasoning tasks.(3) *EMA Sensitivity:* The performance is sensitive to the choice of the EMA momentum parameter. While this reflects the delicate balance required for stable training, it also highlights an opportunity to develop adaptive or learned momentum strategies to improve robustness. These considerations represent opportunities to build upon OSKAR's significant progress and extend its capabilities even further.

**Acknowledgment:** This research is funded by the Swiss National Science Foundation (SNSF) through the project grant:10003100. Computational resources were provided as part of the Swiss AI Initiative by a grant from the Swiss National Supercomputing Centre (CSCS) under project ID a03 on Alps.

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
