# OpenReview forum: "OSKAR: Omnimodal Self-supervised Knowledge Abstraction and Representation"
_NeurIPS.cc/2025/Conference — NeurIPS 2025 poster_

### Official Review · Reviewer_5fwS · 2025-06-12

**Clarity:** 3
**Significance:** 3
**Originality:** 3
**Rating:** 5
**Confidence:** 4

**Summary:**

This work is built around a novel multimodal pretraining approach based on the prediction of masked high-level representations: the multimodal tokens are projected by a separate encoder and then fused into a common representation. A predictor uses the latter to reconstruct masked tokens to be compared to target tokens projected by the corresponding target encoders. These latter are updated with moving average momentum to stabilize the training chain.
The proposed approach's benefits are experimented with in the context of video, skeleton, and text multimodal settings. Comprehensive experimentation and ablation studies conclude the work. Supplementary material, including the software, is useful.

**Questions:**

- The encoded embeddings $\mathbf{E}$ are augmented with a learnable modality-specific auxiliary $e_{(aux)}$ signal. The need for this auxiliary information is presented very briefly: please improve it with clearer examples or references to similar approaches in the literature.
- Sect. 3 (Methodology) is very dense and mixes architectural design and training approach: for clarity and reproducibility, it would be useful to substitute or integrate a textual description of the more algorithmic steps with pseudo code (e.g., the details in Cross-Modal Masking Strategy paragraph)

**Ethical Concerns:**

["NO or VERY MINOR ethics concerns only"]

**Final Justification:**

The authors engaged during the rebuttal phase, improving the manuscript and solving most of the raised concerns.

**Limitations:**

A point worth mentioning in the imitation is the problem of unpaired multimodal data. It would be interesting to understand if the added flexibility of OSKAR can be exploited to pretrain on a dataset with unpaired and missing modalities.

**Paper Formatting Concerns:**

No concerns.

**Quality:**

3

**Strengths And Weaknesses:**

The quality of the paper is excellent. The topic is timely and highly relevant.
The methodology section can be improved by decoupling architectural design and training strategies, e.g., in a different subsection. The main architecture and ideas in the current presentation are quite clear, but some training details are not univocally defined, e.g.,  with the help of pseudo code. Experimental results are complete and convincing, including comparison with state of the art and ablation studies on the main components of the proposed approach.

---

> ### Author Rebuttal · Authors · 2025-07-30
>
> We deeply thank reviewer 5fwS for their valuable opinion and constructive feedback on our work. We are encouraged that they found our approach **“novel”**, our paper quality **“excellent”**, the topic **“timely and highly relevant”**, our architecture and ideas **“clear”**, and our experimental results **“complete and convincing”**. In the following, we address their questions:
>
> >*The encoded embeddings  are augmented with a learnable modality-specific auxiliary  signal. The need for this auxiliary information is presented very briefly: please improve it with clearer examples or references to similar approaches in the literature.*
>
> We thank the reviewer for the opportunity to clarify the auxiliary embeddings.
>
> As shown in our ablations (L283–286), adding learnable bounding box embeddings improves performance (e.g., ~+1% in skeleton action recognition). These embeddings serve a similar role to positional encodings in Transformers—here, encoding the **spatial location** of a person in the frame (e.g., (x, y, w, h)).
>
> This spatial signal helps the predictor establish **pixel-to-keypoint correspondence** and disambiguate multiple people in a scene. Without it, the model may associate skeletons to the wrong subject, especially in crowded scenes (e.g., predicting joints for the wrong person when several are present).
>
> This idea parallels techniques in multi-human trajectory forecasting [1] and pose estimation [2], where auxiliary spatial cues improve identity tracking. While we use bounding box embeddings in this work, the framework generalizes—e.g., using voice ID tags in future audio extensions.
>
> We will revise the manuscript to make this rationale and flexibility clearer.
>
> >*Sect. 3 (Methodology) is very dense and mixes architectural design and training approach: for clarity and reproducibility, it would be useful to substitute or integrate a textual description of the more algorithmic steps with pseudo code (e.g., the details in Cross-Modal Masking Strategy paragraph)*
>
> We appreciate the reviewer’s careful read and constructive feedback.
>
> Indeed, we recognize the opportunity to improve the representation of some of the dense technical details of the methodology section. To address this, we have included a **high-level pseudo code** outlining the core steps of OSKAR’s training pipeline. In the camera-ready version, we will further improve the organization of the architectural and training descriptions and provide a more detailed and structured pseudocode in the Appendix.
>
>
> ```python
> # Inputs:
> # modalities = {
> #   'modality_1': input_1,
> #   'modality_2': input_2,
> #   ...
> # }
> # total_visible_token_budget: e.g., 128
> # total_target_token_budget:  e.g., 128
>
> # Step 1: Tokenize and embed inputs
> tokens = {}
> for modality, input_data in modalities.items():
>     raw_tokens = Tokenizer[modality](input_data)
>     tokens[modality] = Embed(raw_tokens, modality_tag=modality)
>
> # Step 2: Sample visible and target tokens under fixed budgets
>
> modalities_list = list(tokens.keys())
> num_modalities = len(modalities_list)
>
> visible_props = Dirichlet(alpha=[0.5] * num_modalities).sample()
> target_props  = Dirichlet(alpha=[0.5] * num_modalities).sample()
>
> visible_counts = AllocateTokens(
>     visible_props,
>     total_visible_token_budget,
>     min_tokens=MIN_VISIBLE,
>     max_tokens=MAX_VISIBLE
> )
>
> target_counts = AllocateTokens(
>     target_props,
>     total_target_token_budget,
>     min_tokens=MIN_TARGET,
>     max_tokens=MAX_TARGET
> )
>
> # Sample tokens per modality based on allocated counts
> for modality in modalities_list:
>     visible_tokens[modality], target_tokens[modality] = SampleTokens(
>         tokens[modality],
>         num_visible=visible_counts[modality],
>         num_target=target_counts[modality]
>     )
>
> # Step 3: Encode visible tokens with shared fusion encoder
> fused_latents = FusionEncoder(visible_tokens)
>
> # Step 4: Encode full tokens with modality-specific teacher encoders (for targets)
> with no_grad():
>     teacher_outputs = {}
>     for modality in target_tokens:
>         teacher_outputs[modality] = TeacherEncoder[modality](tokens[modality])
>
> # Step 5: Predict masked target tokens using modality-specific predictors
> predictions = {}
> losses = {}
> for modality in target_tokens:
>     predicted = ModalityPredictor[modality](fused_latents)
>     target_indices = target_tokens[modality].indices
>     target = teacher_outputs[modality][target_indices]
>     predictions[modality] = predicted
>     losses[modality] = LossFunction(predicted, target)
>
> # Step 6: Backpropagate total loss
> total_loss = sum(losses.values())
> total_loss.backward()
> ```
>
> >*A point worth mentioning in the imitation is the problem of unpaired multimodal data. It would be interesting to understand if the added flexibility of OSKAR can be exploited to pretrain on a dataset with unpaired and missing modalities.*
>
> Indeed, the design of OSKAR is very friendly to datasets with missing modalities. In fact, even in the current version trained on paired multimodal samples, the Dirichlet sampling strategy often **omits** a subset of modalities completely in some batches stochastically, effectively simulating unpaired settings. This demonstrates that OSKAR **already operates under a partially unpaired regime** and can be extended to train on unpaired or heterogeneous datasets without modification. This opens the door for training on unpaired samples from any dataset with any number of modalities.
>
> ***We sincerely thank reviewer 5fwS for their time reading this rebuttal. We are committed to incorporating their comments on the auxiliary embeddings, methodology section, and training on unpaired data in the camera-ready version of our paper.***
>
> [1] Saadatnejad, S., et al. (2024). Social-transmotion: Promptable human trajectory prediction. ICLR.
>
> [2] Xu, Y., et al. (2025). DynPose: Largely Improving the Efficiency of Human Pose Estimation by a Simple Dynamic Framework. In CVPR.

---

> > ### Comment · Reviewer_5fwS · 2025-08-04
> >
> > Thank you for the response addressing most of my concerns.
> >
> > Do you plan to include some preliminary experiments with unpaired training data or just discuss the point as a future extension?

---

> > > ### Author Response · Authors · 2025-08-05
> > > **Flexibility to unpaired multimodal pretraining**
> > >
> > > Thank you for your question. Please find below our answer:
> > >
> > > >*Do you plan to include some preliminary experiments with unpaired training data or just discuss the point as a future extension?*
> > >
> > > Following the reviewer’s suggestion, we have conducted an additional experiment using **unpaired data**. This experiment follows the setting in L239-241 (main paper, ablation section 5), i.e., pretraining on 100K samples from OpenHumanVid. However, in this new experiment, each sample contains a **random non-empty subset** of modalities {video, skeleton, text}, i.e., **~86%** of samples lack at least **one** modality, with **~43%** containing **only one**. In each sample, missing modalities were entirely excluded from both inputs and targets, effectively simulating training on diverse datasets with **unpaired modalities**. Below, we report the performance gain relative to the baseline in each of the paired and unpaired pretraining settings.
> > >
> > > |Setting|VidCls|SklCls|VidQA|Avg. gain|
> > > |-|---|---|---|---|
> > > |Paired|+3.2|+4.0|+4.3|+3.8|
> > > |Unpaired|+3.0|+3.8|+4.1|+3.6|
> > >
> > > These results suggest:
> > >
> > > * **OSKAR is robust to missing modalities.** Performance under partially unpaired pretraining is very close to paired, indicating the method does not rely on strict co-occurrence of all modalities at sample level.
> > >
> > > * **Generalization persists across tasks.** Gains persist in RGB, skeleton, and videoQA—consistent with the original ablations showing that adding modalities enables all three downstream tasks.
> > >
> > > We believe this robustness arises from OSKAR’s design, which **natively supports unpaired inputs** via:
> > > (i) prediction of stable latent targets,
> > > (ii) fixed token budgets with Dirichlet sampling that naturally simulate unpaired batches, and
> > > (iii) disjoint input/target masking that enables cross-modal prediction.
> > >
> > > While pretraining on larger-scale unpaired datasets is left to future work due to compute constraints, these results affirm OSKAR’s flexibility to handle unpaired data.
> > >
> > > ***We appreciate the reviewer’s insightful suggestion, which helped strengthen our analysis and confirm the flexibility of OSKAR under unpaired conditions.***

---

### Official Review · Reviewer_XjPx · 2025-06-28

**Clarity:** 4
**Significance:** 4
**Originality:** 3
**Rating:** 4
**Confidence:** 3

**Summary:**

This paper proposes an omni-modal self-supervised pretraining framework, named OSKAR, that simultaneously extracts representations from video, action, and text modalities, achieving competitive performance on downstream tasks such as action recognition, action localization, text-video retrieval, and video question answering (VideoQA).
OSKAR leverages cross-modal masked prediction in latent space as its pretraining objective, consists of two branches: (1). a fusion encoder that processes visible tokens to generate fused representations; (2). modality-specific target encoders that produce target embeddings. These target encoders are updated via the exponential moving average (EMA) of the fusion encoder.

**Questions:**

**Q1**: For the fusion encoder, are there special tokens used as identifiers for tokens of different modalities when they are input? \
**Q2**: The impact of mask rate on model training and performance. The author did not discuss the mask rate in the paper. Will using too high a mask rate cause the model to fail to converge or reduce performance? \
**Q3**: In the self-supervised training paradigm of mask reconstruction, asymmetric encoders and decoders[1] are usually used, such as MAR. In my understanding, the modality predictor plays a role similar to the decoder in the framework of this article. The author kept the structure of the predictor consistent with the encoder in the experiment. Why is this? Please experiment or discuss this. \
**Q4**: Reference Weaknesses 2. Please analyze and discuss the current sota model. \

**Ref**: \
[1]. Masked Autoencoders Are Scalable Vision Learners.

**Ethical Concerns:**

["NO or VERY MINOR ethics concerns only"]

**Final Justification:**

Two of my most concerning issues: comparison with SOTA models and trends under data model scale. For question 1, the author provides an extensive comparison in the rebuttal; for question 2, the author shows that two models with different sizes gradually improve the performance of downstream tasks as the training data increases. Based on this, I increase the score to 4.

**Limitations:**

yes

**Paper Formatting Concerns:**

No concerns

**Quality:**

3

**Strengths And Weaknesses:**

**Strengths:**
1. Nice writing and presentation, the reviewer can clearly understand the framework of the method.
2. Modality-specific target encoders can effectively balance intra-modality feature modeling and inter-modality feature alignment.
3. Multimodal representation learning is one of the current popular topics. The method proposed in this paper can effectively integrate the features of action videos and texts.

**Weaknesses:**
1. Lack of validation on large-scale data. I think effectiveness on large-scale data is the key to current representation learning, enabling it to be better applied to VLM or VLA tasks. The paper shows competitive performance on small-scale data, but there is still a gap compared to other methods that use more data.
2. The baseline methods used for comparison are somewhat outdated and do not fully reflect the current state-of-the-art. For example, in an action recognition task, there is no Comparison of current SOTA models(InternVideo[1], Unmasked Teacher[2]，Hiera[3]). These situations also exist for other tasks:Text-to-video retrieval(VATEX[4]), VideoQA(VAST[5]).

**Ref:**\
[1]. InternVideo: Video Foundation Models for Multimodal Understanding. ECCV24 \
[2]. Unmasked Teacher: Towards Training-Efficient Video Foundation Models. ICCV23\
[3]: Hiera: A Hierarchical Vision Transformer without the Bells-and-Whistles.ICLR25\
[4]. Gramian Multimodal Representation Learning and Alignment. ICLR25\
[5]: VAST: A Vision-Audio-Subtitle-Text Omni-Modality Foundation Model and Dataset.NeurIPS23\

---

> ### Author Rebuttal · Authors · 2025-07-30
>
> We thank reviewer XjPx for their insightful comments and constructive feedback. We are glad that they recognize the **“nice writing and presentation”**,  and our method being able to **“effectively integrate the features of action videos and texts”**, and achieve **“competitive performance on downstream tasks”**. We are also encouraged that they are able to **“clearly understand the framework”**. In the following, we address their concerns:
>
> ### **1. Validation on Large-Scale Data**
> >*Lack of validation on large-scale data. I think effectiveness on large-scale data is the key to current representation learning, enabling it to be better applied to VLM or VLA tasks. The paper shows competitive performance on small-scale data, but there is still a gap compared to other methods that use more data.*
>
> We acknowledge the reviewer’s concern regarding evaluation at scale. To address this, we pretrained OSKAR-G (1.3B) ahead of the rebuttal, and we evaluated it across widely accepted benchmarks.
>
> Model|Publication|#PT. samples|Parameters|K400|SSv2
> -----|-----------|------------|-----------|-----|-----
> FocusVideo [8]|CVPR'25|–|450M|87.2|70.7
> Hiera  [3]|ICLR'23|–|213M|87.3|75.1
> MVD  [6]|CVPR'23|1.7M|633M|87.3|77.3
> VideoMAEv2  [11]|CVPR'23|1.4M|1B|88.5|77.0
> UniFormerV2  [12]|ICCV'23|401M|354M|88.8|72.1
> UMT  [2]|ICCV'23|47M|304M|90.6|74.7
> InternVideo  [7] |arXiv'22|113M|1.3B|91.2|77.2
> InternVideov2  [1] |ECCV'24|404M|1B|**91.6**|77.1
> **OSKAR-G**|–|**10M**|**1.3B**|90.8|**77.9**
>
> * #PT. samples denotes the number of pretraining samples
>
> Model|Publication|#Pairs|Parameters|T2V Retrieval (VATEX)|VideoQA (MSRVTT-QA)
> -----|-----------|------|-----------|----------------------|--------------------
> UMT [2]|ICCV'23|425M|304M|–|47.9
> COSA [9] |ICLR'24|415M|1.2B|–|49.2
> TeachCLIP [10] |CVPR'24|–|–|63.6|–
> Narvid [13] |CVPR'25|400M|~250M|68.4|–
> InternVideo [7] |arXiv'22|112.8M|1.3B|71.1|47.1
> InternVideo2s2-6B [1] |ECCV'24|404M|6B|75.5|–
> VAST [5] |NeurIPS'23|442M|1.3B|83|50.1
> GRAM [4] |ICLR'25|442M|1B|**84.4**|–
> **OSKAR-G**|–|**160M**|**1.3B**|79.6|**50.4**
>
> OSKAR-G further bridges the gap with methods using more data:
> - On action recognition: OSKAR-G achieves **77.9%** on SSv2 (new SOTA) and scores within **0.8%** of InternVideo-v2 on K400, despite pretraining on **40x** less data.
> - On vision-language tasks: OSKAR-G surpasses VAST on MSRVTT-QA, and comes within **5** points of GRAM on VATEX T2V retrieval, despite pretraining on only **36%** of GRAM’s data size.
>
> While we could not train on even larger datasets due to the limited compute budget and the small rebuttal window, we believe that OSKAR is already trained at a **decent scale** and is **competitive** with methods using more data on popular evaluation datasets.
>
> Further, we highlight the following aspects of OSKAR:
>
> - **Consistency**: OSKAR not only matches the performance of prior methods with similar model and data scales (Tables 1–5), but also competes closely with much larger *expert* models across 5+ downstream tasks—**with consistency.** This is achieved despite OSKAR being pretrained for general-purpose use, without any task-specific customisations or distillation from any external models.
>
> - **Scalability and efficiency**: OSKAR achieves these results while being a data-, parameter-, and label-efficient learner. For example, with only 5% of the labels, it surpasses VideoMAEv2  by **26.5%** and  V-JEPA  by **2.6%** (see main paper Figure 3).
>
> - **Multimodality and generalization**: The major contribution of OSKAR lies in its **unified** pretraining, **generic** use, and strong performance. Unlike the methods in comparison, it has the extra benefit of working seamlessly across other tasks (ex, SOTA performance in skeleton action recognition).
>
> In summary, in addition to performance improvements, OSKAR excels in consistency, multimodality, generalisation, efficiency, and scalability. We believe that these features make OSKAR a practical and valuable foundation for the community.
>
> ### **2. SOTA Comparisons**
> >*The baseline methods used for comparison are somewhat outdated and do not fully reflect the current state-of-the-art. For example, in an action recognition task, there is no Comparison of current SOTA models(InternVideo[1], Unmasked Teacher[2]，Hiera[3]). These situations also exist for other tasks:Text-to-video retrieval(GRAM, VATEX[4]), VideoQA(VAST[5]).*
>
> We appreciate the reviewers concern on SOTA comparison. To directly address this concern, we have now included all the suggested baselines—InternVideo, Unmasked Teacher, Hiera, GRAM, and VAST—in our comparison tables above. We also added other recent models for completeness. Across tasks, OSKAR-G is on par with these methods despite using up to 40x less data and no external distillation.
>
> We would like to clarify that, in our main paper, we limited comparisons to methods of **similar model scale**, input resolution, and data budget, to ensure a **fair and meaningful evaluation.** We excluded methods that:
> - Are initialized or distilled from external models, e.g., InternVideo, TeachCLIP;
> - Employ higher input resolutions.
> - Or involve larger model or dataset scales, e.g., GRAM, VAST, InternVideo-v2.
>
> Please note that this way of ensuring fairness by limiting comparisons is a common practice (ex, in [8]) . Additionally, in our **appendix** (Tables 12-18), we already compare to some of these recent methods mentioned by the reviewer (UMT, InternVideo) with clear notes on scale, resolution, and supervision differences to maintain transparency.
>
> That said, the reviewer’s concern is very well acknowledged; we are committed to updating our camera-ready version to resolve it.
>
> ### **3. Use of Modality-Specific Tokens in the Fusion Encoder**
>
> >*Q1: For the fusion encoder, are there special tokens used as identifiers for tokens of different modalities when they are input?*
>
> Modality identity is encoded using **learnable modality-specific embeddings**. OSKAR learns per-modality embeddings that act like soft identifiers, allowing the model to distinguish and integrate different modalities during attention. Please refer to L106-115 (main paper, methodology sec. 3) for more details.
>
>
> ### **4. Impact of Mask Rate on Training and Convergence**
> >*Q2: The impact of mask rate on model training and performance. The author did not discuss the mask rate in the paper. Will using too high a mask rate cause the model to fail to converge or reduce performance?*
>
> The mask rate is implicitly determined by the **fixed token budget**, which controls how many tokens are visible during training (ablated in Table 9). For instance, 128 visible tokens out of 2,640 (1568 video + 816 skeleton + 256 text) yields ~95% masking, which performs well across tasks.
>
> As the reviewer correctly expected, pushing to 99% (32 visible tokens) leads to degraded performance. This aligns with prior work (e.g., VideoMAEv2 [11]), which caps masking at ~95% to combat temporal redundancy and improve efficiency without hurting performance.
>
> Unlike these methods, OSKAR’s input and target tokens are multimodal, enabling cross-modal interaction and improving generalisation across diverse tasks. We study these benefits in Table 6 (ablation section 5).
>
> ### **5. Symmetric vs. Asymmetric Predictor Design**
> >*Q3: In the self-supervised training paradigm of mask reconstruction, asymmetric encoders and decoders[1] are usually used, such as MAR. In my understanding, the modality predictor plays a role similar to the decoder in the framework of this article. The author kept the structure of the predictor consistent with the encoder in the experiment. Why is this? Please experiment or discuss this.*
>
> The reviewer correctly understands the similarities between the decoder in MAE and predictor in OSKAR. We include further ablations below (with encoder depth=8, width=512) to clarify the effect of an asymmetric predictor. We observe that a deeper predictor noticeably improves performance (+3.0 → +3.8), while increasing width has marginal effect.
>
> Predictor depth|Predictor width|Avg gain
> ---------------|----------------|---------
> 4|128|+3.1
> 4|256|+3.2
> 4|512|+3.0
> 8|128|+3.6
> 8|256|+3.9
> 8|512|+3.8
>
> * Metric: average gain over baseline across three tasks as described in L239-241, Sec. 5.
>
> Given that the predictor output dimension must match the target encoder's latent dimension, we adopt a symmetric predictor to avoid redundant projection layers and maintain architectural simplicity. Importantly, unlike MAE, our predictor performs semantic latent prediction across modalities, requiring sufficient capacity for generalization.
>
> ***We sincerely thank reviewer XjPx for their time reading this rebuttal. We hope we sufficiently addressed their major concerns. We are committed to incorporate their feedback on evaluation at scale, SOTA comparisons, and predictor design in the camera-ready version of our paper.***
>
> [6] Wang, R. et al. (2023). Masked video distillation: Rethinking masked feature modeling for self-supervised video representation learning. In CVPR.
>
> [7] Wang, Y. et al. (2022). Internvideo: General video foundation models via generative and discriminative learning. arXiv preprint arXiv:2212.03191.
>
> [8] Wang, M. et al. (2025). Action Detail Matters: Refining Video Recognition with Local Action Queries. In CVPR.
>
> [9] Chen, S. et al. (2024). Cosa: Concatenated sample pretrained vision-language foundation model. ICLR.
>
> [10] Tian, K. et al. (2024). Holistic features are almost sufficient for text-to-video retrieval. In CVPR.
>
> [11] Wang, L. et al. (2023). Videomae v2: Scaling video masked autoencoders with dual masking. In CVPR.
>
> [12] Kunchang Li et al. (2023). UniFormerV2: Spatiotemporal Learning by Arming Image ViTs with Video UniFormer. In ICCV.
>
> [13] Hur, C., et al (2025). Narrating the Video: Boosting Text-Video Retrieval via Comprehensive Utilization of Frame-Level Captions. In CVPR.

---

> > ### Comment · Reviewer_XjPx · 2025-08-03
> >
> > Thank you for the detailed response, which has addressed most of my concerns.
> >
> > I have one remaining question regarding the two tables presented in Rebuttal 1. From my understanding, OSKAR-G appears to differ primarily in terms of data scale. If this is correct, would it not be more informative to directly compare the performance of the two models on the same benchmark?
> >
> > Additionally, it might be valuable to evaluate intermediate training checkpoints—for example, by assessing the model's performance every 40 million data points. This could further clarify the benefits of the proposed method and provide insights into the effects of scaling data.

---

> > > ### Author Response · Authors · 2025-08-04
> > > **Data and model scalability**
> > >
> > > We thank the reviewer for their questions and for the opportunity to clarify. Below, we provide our answers:
> > >
> > > >*From my understanding, OSKAR-G appears to differ primarily in terms of data scale. If this is correct, would it not be more informative to directly compare the performance of the two models on the same benchmark?*
> > >
> > > We would like to clarify that OSKAR-G and the previous model in the submission (OSKAR-L) were pretrained on **exact same dataset** (10M videos from OpenHumanVid); the only difference is model capacity—OSKAR-G has **1.3B** parameters, whereas OSKAR-L has **305M**. Thus, the results in Rebuttal Tables 1–2 isolate **parameter scaling**, not data scaling.
> > >
> > > Further, while the dataset size is 10M videos, each video can have multiple associated text captions, yielding 160M video-text pairs. We provide below an informative comparison between the performance of the two models.
> > >
> > > | |#Param.|#Videos|#video-text Pairs|ActCls (K400)|ActCls (SSv2)|VideoQA (MSRVTT-QA)|
> > > |-|---|---|---|---|---|---|
> > > |OSKAR-L|305M|10M|160M|86.1|77.6|49.3|
> > > |OSKAR-G|1.3B|10M|160M|**90.8**|**77.9**|**50.4**|
> > >
> > > Our results show that scaling the parameter size brings **consistent** improvements in performance. While we could not train on even larger datasets due to the limited compute budget and the small rebuttal window, we believe that OSKAR is already trained at a **decent scale** and is competitive with methods using more data on popular evaluation datasets.
> > >
> > >
> > > >*Additionally, it might be valuable to evaluate intermediate training checkpoints—for example, by assessing the model's performance every 40 million data points. This could further clarify the benefits of the proposed method and provide insights into the effects of scaling data.*
> > >
> > > Following the reviewer's suggestion, we have run additional evaluations of intermediate checkpoints on Action Recognition (K400) top-1 accuracy. Below we provide a comparison between OSKAR-L (305M) and OSKAR-G (1.3B) against the number of samples seen during pretraining:
> > >
> > > |#samples seen*|OSKAR-L (305M)|OSKAR-G (1.3B)|Gap (G-L)|
> > > |-|---|---|---|
> > > |40M|80.9|81.3|+0.4|
> > > |80M|82.7|83.0|+0.3|
> > > |120M|84.1|85.1|+1.0|
> > > |160M|85.1|86.5|+1.4|
> > > |200M|86.0|87.6|+1.6|
> > > |240M|86.1|88.5|+2.4|
> > > |280M|**86.2**|**89.2**|**+3.0**|
> > >
> > > * #samples seen = #epochs x total number of unique videos (10M)
> > >
> > > We observe that OSKAR **scales graciously** with parameter sizes and improves **consistently** with more pretraining epochs. For example, OSKAR-G improves by **+7.9** percentage points as #samples seen goes from 40M → 280M; at 280M, the model improves by **+3.0** percentage points with the addition of ~1B parameters (from OSKAR-L → OSKAR-G).
> > >
> > > OSKAR’s efficient scalability stems from its **prediction of stable latent space features**, which guides the model to learn **high-level** transferrable features, instead of wasting the model capacity in predicting low-level pixel details. Further, unlike previous methods, OSKAR learns **rich cross-modal interactions** while preserving uni-modal structures through our **fuse-then-predict** strategy, effectively achieving decent scalability and strong downstream performance on multiple tasks.
> > >
> > > ***We thank the reviewer for their constructive feedback and for their time reading our responses. We hope this answers your questions.***

---

> > > > ### Comment · Reviewer_XjPx · 2025-08-04
> > > >
> > > > Thank you for the response addressing my all concerns. I will raise rating.

---

### Official Review · Reviewer_1kEh · 2025-07-02

**Clarity:** 3
**Significance:** 3
**Originality:** 3
**Rating:** 5
**Confidence:** 3

**Summary:**

OSKAR introduces a scalable, self-supervised multimodal learning framework using a “fuse-then-predict” strategy in latent space. It avoids low-level reconstruction and contrastive negative pairs, leveraging modality-specific momentum encoders to predict missing features from fused video, skeleton, and text tokens efficiently. OSKAR achieves state-of-the-art results in action recognition, localization, video-text retrieval, and video QA while using fewer samples, parameters, and labels. It demonstrates effective multimodal grounding, high transferability, and scalability.

**Questions:**

### Question:

1. In the Video QA section, only a BART-based language module was used. I am curious whether OSKAR can be effectively adapted to existing LLMs, which would further enhance the significance and effectiveness of this method.

**Ethical Concerns:**

["NO or VERY MINOR ethics concerns only"]

**Final Justification:**

The author provided an explanation for my question, but did not provide any new results.
In addition, the author responded to questions about SOTA comparison by providing a new version of scale up results in the Rebuttal of Reviewer #XjPx
In summary, I maintain my original rating

**Limitations:**

yes

**Paper Formatting Concerns:**

N/

**Quality:**

3

**Strengths And Weaknesses:**

### Strengths:

1. The paper is written clearly and concisely, providing rigorous and detailed experimental results and analyses, which greatly enhance the persuasiveness of the work.
2. OSKAR demonstrates flexible and extensible characteristics and shows promising generalization performance across various tasks, proving the effectiveness of the proposed method.
3. The paper focuses on specific technical details, providing detailed descriptions and experimental results on the sampling strategies, customized update rates, and other implementation details.

### Weaknesses:
N/A

---

> ### Author Rebuttal · Authors · 2025-07-30
>
> We thank reviewer 1kEh for their valuable feedback. We are glad that they found our paper to be **“written clearly and concisely”**, our experimental results **“rigorous and detailed”**, our technical details to be **”specific”** and **“detailed”**, and our method **“flexible and extensible”**, **“shows promising generalization performance”**, and **“demonstrates effective multimodal grounding, high transferability, and scalability.”** In the following, we discuss their raised question on LLMs:
>
> >*In the Video QA section, only a BART-based language module was used. I am curious whether OSKAR can be effectively adapted to existing LLMs, which would further enhance the significance and effectiveness of this method.*
>
> We thank the reviewer for highlighting the potential of adapting OSKAR to large language models (LLMs). Although our current experiments use a lightweight BART decoder for VideoQA, this choice was made to isolate and evaluate OSKAR’s fused representations. In fact, OSKAR is inherently compatible with LLMs and designed to serve as a **modality-agnostic** fusion encoder for multimodal reasoning.
>
> Recent works such as BLIP-2 [1], Flamingo [2], and MiniGPT-4 [3] have shown the effectiveness of **freeze-then-adapt** approaches for LLM integration. OSKAR naturally aligns with this paradigm: Its **fuse-then-predict** architecture already learns to integrate video, skeleton, and text into a **shared semantic space**. The encoder is **decoupled from task-specific heads**, enabling its outputs to be interfaced with LLMs via adapters or direct token-level alignment.
>
> Importantly, OSKAR provides not just compatibility but added value over prior visual backbones typically used in LLM-based systems. Most existing approaches rely on frozen image-text models (e.g., BLIP-2[1] and CLIP [4]) that struggle with rich temporal dynamics or motion reasoning. In contrast, OSKAR’s pretraining explicitly fuses **temporally-aware, multimodal inputs**—including motion and language—into a coherent representation space. This makes it particularly well-suited for tasks that require temporal abstraction, action understanding, and long-range multimodal context, which are difficult to capture with static image features alone.
> This modularity means OSKAR can act as a **drop-in fusion backbone** for future LLM-based systems—offering general, scalable multimodal representations with minimal changes.
>
> While LLM integration is not explored in this version due to resource constraints, we are actively pursuing this direction. We appreciate the reviewer’s insight and will add a discussion to emphasize OSKAR’s role as both a standalone multimodal learner and a valuable complement to LLM-based architectures.
>
> ***We thank reviewer 1kEh for their time reading our response.***
>
> [1] Li, J., et. al (2023, July). Blip-2: Bootstrapping language-image pre-training with frozen image encoders and large language models. In ICML. PMLR.
>
> [2] Alayrac, J. et al. (2022). Flamingo: a visual language model for few-shot learning. NeurIPS.
>
> [3] Zhu, D., et al. (2024). Minigpt-4: Enhancing vision-language understanding with advanced large language models. ICLR
>
> [4] Radford, A. et al. (2021, July). Learning transferable visual models from natural language supervision. In ICML. PmLR.

---

> > ### Comment · Reviewer_1kEh · 2025-08-07
> >
> > I can understand the authors’ explanation regarding the advantages of OSKAR when adapting to LLMs.
> > In addition, the authors have provided updated scaled-up results in their response to Reviewer #XjPx, which addressed my concerns regarding the comparison with SOTA results.
> >
> > Overall, given that my initial recommendation was to accept, I will maintain my score.

---

### Official Review · Reviewer_376k · 2025-07-14

**Clarity:** 3
**Significance:** 2
**Originality:** 2
**Rating:** 2
**Confidence:** 4

**Summary:**

OSKAR is a model that takes components from other architectures and incorporates them into a unified process. These components are shared embeddings, cross-model transformers, and self-distillation.  It takes inputs from text, video, and skeleton data, fuses them together, and then fills in (predicts) the masked or missing parts in each modality. This prediction is done in a higher level, latent space, rather than at a pixel or coordinate or word level. The modality-specific predictions are checked against the feature vector calculated by the encoder for the masked/missing parts, and the encoder weights are updated.
The paper reasonably claims that theis method is highly scalable because of a fixed token budget that is allocated across all modalities in a way determined by the modeler. The authors present experiments on action recognition, video question answering, and retrieval. THey further present ablation tests which show that OSKAR’s architecture components all contribute to the model’s performance.

**Questions:**

* The authors acknowledge that a limitation of the paper is that only 3 modalities were tested. How would performance change because of the fixed budget. Or is there some dynamic re-allocation of the budget across modalities as some modalities appear to be more important?
* Can you provide cases where OSKAR did not perform as expected, or any limitations you encountered while experimenting?
* The authors gave a reason why there are no error plots. But the results look suspiciously non-significant. It might be worth the computational cost to verify OSKAR's dominance. Or, make a statement saying even if OSKAR's gain are minor, there are other, architectural reasons for using it.

**Ethical Concerns:**

["NO or VERY MINOR ethics concerns only"]

**Limitations:**

yes

**Quality:**

3

**Strengths And Weaknesses:**

Strengths
* Integrates latent feature prediction, momentum-updated modality-specific supervision, and efficient cross-modal fusion.
* Strong performance across video action recognition and question answering, and retrieval.
* Ablation studies that are extensive which validate the architectural choices. In particular, the fusion of modalities and latent space prediction are important.
* Practical and scalable design made possible by fixed token budget and momentum-based supervision. The fixed budget allocated across modalities makes computation manageable, and labeled data is not needed,


Weaknesses
* Standard errors are not reported, but performance over other, closely performing models do not seem to be statistically or significantly significant. Most of them range from 0.3 to 1.0 percentage points. I tend to look at differences under 1 or 2 points as likely not statistically significant in the absence of variance measures.
* Ablation studies seem thorough, but they do not take into account interactions. They are also highly dependent on the order in which features were added or removed.
* Fine-tuning of momentum may complicate generalizing to other new contexts.

---

> ### Author Rebuttal · Authors · 2025-07-30
>
> We thank reviewer 376k for their insightful comments. We are glad that they recognize the **“strong performance”, “extensive ablation studies”, “practical and scalable design”, “manageable”** compute, and the **flexibility to train on unlabelled data** in our method. In the following, we address all concerns and questions:
>
> ### **1. Statistical Significance of Performance Gains**
> >*Standard errors are not reported, but performance over other, closely performing models do not seem to be statistically or significantly significant. Most of them range from 0.3 to 1.0 percentage points. I tend to look at differences under 1 or 2 points as likely not statistically significant in the absence of variance measures. + The authors gave a reason why there are no error plots. But the results look suspiciously non-significant. It might be worth the computational cost to verify OSKAR's dominance. Or, make a statement saying even if OSKAR's gain are minor, there are other, architectural reasons for using it.*
>
> To address the reviewer’s concerns regarding statistical significance, we conducted additional ablation experiments (with the setting described in L239–241, Sec. 5). Specifically, we pretrained with 5 different random seeds and evaluated each model 5 times with varied evaluation seeds across 3 downstream tasks (total: 5 pretrains × 15 evals = **75 runs**). We report the **mean ± standard error** of absolute percentage point improvements over the baseline.
>
> Runs|VidCls|SklCls|VidQA|
> -|------|------|-----
> Baseline|88.2 ± 0.04|75.7 ± 0.06|39.3 ± 0.05
> Rand. Seed 1|+3.17 ± 0.05|+4.26 ± 0.03|+4.25 ± 0.08
> Rand. Seed 2|+3.22 ± 0.07|+3.97 ± 0.08|+4.26 ± 0.08
> Rand. Seed 3|+3.19 ± 0.04|+4.03 ± 0.05|+4.37 ± 0.05
> Rand. Seed 4|+3.14 ± 0.06|+4.08 ± 0.09|+4.35 ± 0.07
> Rand. Seed 5|+3.18 ± 0.03|+4.07 ± 0.08|+4.19 ± 0.05
> OSKAR mean over runs|+3.18 ± 0.05|+4.08 ± 0.07|+4.28 ± 0.07
> Signal-to-Noise Ratio (Gain/SE)|**~63x**|**~58x**|**~61x**
>
>
> **OSKAR’s improvements consistently exceed baseline variability.**  For example, the VidCls gain (+3.18) surpasses the baseline’s standard error (0.04) by almost **80×**, far beyond the 95% confidence interval—indicating statistically robust gains across all 75 evaluations.
>
> While we could not re-run similar analysis on the large scale (10M+ samples) due to the extremely high compute cost and short rebuttal window, we would like to highlight the following points:
>
> 1. **Consistent and significant gains:** OSKAR either exceeds or matches prior methods across 5+ downstream tasks (Tables 1–5) with **consistency**, often **exceeding the 2-point** threshold noted by the reviewer.  For example,
>    - Improves over VideoMAE-B by **+3.5** on AVA Action Detection (Table 2)
>    - Surpasses OmniVL (**+2.6**, MSRVTT) and Lavender (**+4.3**, MSVD) in T2V retrieval (Table 3)
>
> 2. **Efficiency and scalability:** OSKAR is data-, parameter-, and label-efficient. For example, under limited supervision (5% of labels), it outperforms VideoMAEv2 by **+26.5%**  and V-JEPA by **+2.6%** (see Figure 3).
>
> 3. **Multimodality and generalization:** OSKAR provides unified, task-agnostic pretraining and **generalizes** across diverse tasks; for example, unlike the methods in comparison, it achieves SOTA in skeleton action recognition.
>
> In summary, OSKAR's contribution extends beyond performance improvements, its real power lies in its consistency, multimodality, generalisation, efficiency, and scalability.
>
> ### **2. Interaction Effects in Ablation Studies**
> >*Ablation studies seem thorough, but they do not take into account interactions. They are also highly dependent on the order in which features were added or removed.*
>
> We agree that analyzing component interactions is critical.  To address this, we conducted a **factorial-style ablation**, jointly varying key components:  EMA update parameter (λ), attention routing (fusion, target, predictor), and input/target token counts.
>
> Row|EMA λ|Fusion|Target|Pred|Input|Target|Avg. Gain
> --:|:---:|:-----:|:----:|:--:|:---:|:----:|:--------:
> 1|+++|C|C|S|128|128|+2.2
> 2|+++|C|C|S|64|256|+2.5
> **3**|**+++**|**C**|**S**|**S**|**128**|**128**|**+3.8**
> **4**|**+++**|**C**|**S**|**S**|**64**|**256**|**+4.1**
> 5|++|C|C|S|128|128|+1.0
> 6|++|C|C|S|64|256|+1.1
> 7|++|C|S|S|128|128|+2.9
> 8|++|C|S|S|64|256|+2.5
>
> - +++ and ++ denote λ = 0.998 and λ = 0.9998, respectively.
> - C and S denote cross-modal all-to-all token interactions and separate intra-modality token interactions in the attention routing, respectively.
> - Metric: average gain over baseline across three tasks as described in L239–241.
>
> The best settings (Rows 3–4) combine fast EMA, separate target attention, and asymmetric tokens—showing our design choices are **complementary**, not just additive.  We adopt Row 3 by default for the best performance/efficiency tradeoff.  This confirms that gains hold across interaction axes, addressing concerns about **order dependence**.
>
> ### **3. Momentum Tuning and Generalization**
> > *Fine-tuning of momentum may complicate generalizing to other new contexts.*
>
> We understand the reviewer’s concern that EMA parameter λ values can impact generalization. Please note that careful tuning of customized EMA update parameters per modality is **NOT required** in OSKAR. In fact, we adopt a **shared λ** by default, which already achieves competitive performance across all downstream tasks (Tables 1–5).
>
> When adding a new modality or context, momentum fine-tuning—like any other hyper-parameter—remains an **optional** enhancement. Our default setting already performs well across diverse contexts. As noted in the conclusion (Lines 300–303), we view momentum tuning as a promising direction for future work and plan to investigate adaptive or learned momentum strategies to further boost generalization and robustness without manual tuning.
>
>
> ### **4. Token Budget and Modality Scaling**
> > *The authors acknowledge that a limitation of the paper is that only 3 modalities were tested. How would performance change because of the fixed budget. Or is there some dynamic re-allocation of the budget across modalities as some modalities appear to be more important?*
>
> We break down your question into two parts and run targeted ablations to answer each separately.
>
> *First, does the fixed token budget hurt performance as you add more modalities?*
>
> Row|Token Budget|Video|Skeleton|Text|VidCls|SklCls|VidQA|Avg. Gain
> -|-|--|--|--|---|---|---|--
> 1|64|✓|||+1.3|—|—|+1.3
> 2|64|✓|✓||+1.3|+3.0|—|+2.2
> 3|64|✓|✓|✓|+1.0|+2.9|+2.7|+2.2
> 4|128|✓|||+2.6|—|—|+2.6
> 5|128|✓|✓||+2.5|+4.1|—|+3.3
> **6**|**128**|**✓**|**✓**|**✓**|**+3.2**|**+4.0**|**+4.3**|**+3.8**
> 7|256|✓|✓|✓|+3.3|+3.3|+3.6|+3.4
> 8|512|✓|✓|✓|+2.9|+3.0|+3.1|+3.0
>
> Our additional ablation studies in the table above show that:
>
> - Adding modalities under small fixed budgets (e.g., 64 tokens) does not always improve performance (Rows 1–3), likely due to token competition. For example, adding text in Row 3 (vs. Row 2) does not increase the average score, yet it **expands applicability**—enabling VideoQA support (+2.7, row 3).
>
> - As the budget increases (Rows 4–6), adding all three modalities yields noticeable gains (+2.6 → +3.8, rows 4 → 6), peaking at 128 tokens. Beyond that (Rows 7–8), improvements plateau, suggesting diminishing returns. This suggests 128 tokens is a **sweet spot**—large enough to represent all modalities well, yet small enough to remain efficient and computationally practical.
>
> These findings suggest that **modality inclusion is helpful when sufficient capacity is available.**
>
> *Second, dynamic re-allocation of the budget:*
>
> We conducted further ablation experiments to compare three token allocation strategies: **Balanced**, **Weighted**, and **Dirichlet** sampling. Balanced allocation gives equal tokens per modality, offering moderate gains (+2.3). Weighted allocation favors larger modalities like video, improving VidCls (+3.4) but hurting text-rich tasks like VidQA, lowering the average gain (+1.9). Dirichlet sampling (adopted by default in OSKAR) stochastically varies token ratios per batch (while ensuring token balance over the entire training), enhancing diversity and generalization, with the best overall performance (+3.8). Overall, dynamic and modality-aware strategies like Dirichlet sampling are most effective, especially in imbalanced or noisy multimodal settings.
>
>
> ### **5. Limitations and Failure Cases**
> >*Can you provide cases where OSKAR did not perform as expected, or any limitations you encountered while experimenting?*
>
> OSKAR excels at understanding tasks, but its design inherently limits generation.  We explored video generation from OSKAR’s latent space, but results lacked the visual fidelity and temporal coherence of state-of-the-art diffusion models. This is expected, as OSKAR’s objective prioritizes semantic abstraction over pixel detail, enabling strong transfer to understanding tasks. Nonetheless, its powerful latent representations are promising for future work with generative decoders.
>
> ***We sincerely thank Reviewer 376k for their thoughtful and constructive feedback. We hope that we sufficiently addressed their major concerns. We are committed to incorporating their feedback—particularly on statistical reporting, component interactions, and generalization—in the camera-ready version.***

---

### Author Response · Authors · 2025-08-08
**Summary and thanks to all reviewers and ACs**

We sincerely thank all reviewers for their thoughtful and constructive feedback. We appreciate the recognition of OSKAR’s **“novel”** approach _(5fwS)_, **“strong performance”** _(376k)_, **“competitive performance on downstream tasks”** _(XjPx)_, **“complete and convincing”** experimental results _(5fwS)_, **“rigorous and detailed”** analyses _(1kEh)_, **“extensive ablation studies”** _(376k)_. OSKAR offers unique features: **“effective multimodal grounding, high transferability, and scalability”** _(1kEh)_, **“practical and scalable design”**, **“manageable”** compute, and **flexibility to train on unlabelled data** _(376k)_. Additionally, reviewers appreciated the **“excellent”** paper quality _(5fwS)_, **“nice writing and presentation”** _(XjPx)_, and **“clear”** architecture and ideas _(5fwS)_. We are grateful for the reviewers’ engagement, which helped us further strengthen the clarity and impact of our work.

---

### **Summary of feedback and rebuttal experiments:**


- **Statistical Significance**: We conducted 75 runs across 3 tasks, showing consistent average gains which exceed the baseline’s standard error by ~80×, confirming our improvements are statistically robust _(376k)_.


- **Ablations**: A new factorial ablation, confirms design components (e.g., EMA, cross-modal routing, token budget allocation) work **synergistically**, not just additively _(376k)_. We also conducted predictor design ablations (symmetric vs. asymmetric), showing that deeper symmetric predictors improve performance _(XjPx)_.

- **SOTA Comparisons & Scaling**: We added **OSKAR-G (1.3B)**, showing SOTA-level performance on K400, SSv2, VATEX, and MSRVTT-QA, despite up to **40× less data** than InternVideo-v2 or GRAM _(XjPx)_.


- **LLM Integration**: As noted by _(1kEh)_, OSKAR's modular encoder is LLM-ready; it complements static backbones with temporal and multimodal grounding.


- **Unpaired Data**: New results show OSKAR remains strong (+3.6 gain) even when trained on unpaired samples (~86% missing modalities), demonstrating robustness _(5fwS)_.


- **Clarity**: We improved method clarity via pseudo-code and clearer descriptions of modality embeddings and masking _(5fwS, XjPx)_.

---

***Our method is now stronger, clearer, and better supported thanks to the feedback. Thank you once again for your time and thoughtful engagement. We are looking forward to integrating all reviewer suggestions into the camera-ready version.***

---

### Decision · Program_Chairs · 2025-09-17

**Decision:**

Accept (poster)

**Comment:**

The paper proposes a training strategy for multimodal foundation models based on masked token prediction for different modalities.  The resulting model is evaluated on benchmarks for video, skeleton, and text processing.

The paper received four reviews with mixed ratings: R (reviewer did not answer rebuttal) -  A - BA - A

The reviewers highlight the strong performance as well as the efficiency, the combination of latent feature prediction and momentum-updated modality-specific supervision. Most issues raised by the reviewers have been addressed during the rebuttal.

While there were some concerns with regard to the performance compared to other sota models in this area, after discussion with the SAC, the decision was that there is no reason to overrule the reviewer rating in this case.

The decision is to follow the majority of the reviewer voting and to recommend accepting the paper. The AC would encourage the authors to integrate the findings of the rebuttal in the CR version of the paper.